# Wound Healing, Antimicrobial and Antioxidant Properties of *Clinacanthus nutans* (Burm.f.) *Lindau* and *Strobilanthes crispus* (L.) *Blume* Extracts

**DOI:** 10.3390/molecules27051722

**Published:** 2022-03-06

**Authors:** Weng Kit Ban, Isabel Lim Fong, Heng Yen Khong, Joyce Hui Yie Phung

**Affiliations:** 1Department of Paraclinical Sciences, Faculty of Medicine and Health Sciences, Universiti Malaysia Sarawak, Kota Samarahan 94300, Malaysia; wengkitban14@gmail.com (W.K.B.); joycephy24@gmail.com (J.H.Y.P.); 2School of Chemistry and Environment, Faculty of Applied Sciences, Universiti Teknologi MARA, Sarawak Branch, Kota Samarahan 94300, Malaysia; khonghy@uitm.edu.my

**Keywords:** *Clinacanthus nutans*, *Strobilanthes crispus*, wound healing, methicillin-resistant *Staphylococcus aureus*, *Pseudomonas aeruginosa*, scratch assay, antimicrobial, antioxidant

## Abstract

*Clinacanthus nutans* is known to be an anticancer and antiviral agent, and *Strobilanthes crispus* has proven to be an antidiuretic and antidiabetic agent. However, there is a high possibility that these plants possess multiple beneficial properties, such as antimicrobial and wound healing properties. This study aims to assess the wound healing, antioxidant, and antimicrobial properties of *Clinacanthus nutans* and *Strobilanthes crispus*. The *Clinacanthus nutans* and *Strobilanthes crispus* leaves were dried, ground, and extracted with ethanol, acetone, and chloroform through cold maceration. In a modified scratch assay with co-incubation of skin fibroblast and Methicillin-resistant *Staphylococcus aureus*, *Clinacanthus nutans* and *Strobilanthes crispus* extracts were assessed for their wound healing potential, and the antimicrobial activities of *Clinacanthus nutans* and *Strobilanthes crispus* extracts were performed on a panel of Gram-positive and Gram-negative bacteria on Mueller–Hinton agar based on a disc diffusion assay. To assess for antioxidant potential, 2,2-diphenyl-1-picrylhydrazyl (DPPH), total phenolic and total flavonoid assays were conducted. In the modified scratch assay, *Clinacanthus nutans* extracts aided in the wound healing activity while in the presence of MRSA, and *Strobilanthes crispus* extracts were superior in antimicrobial and wound healing activities. In addition, *Strobilanthes crispus* extracts were superior to *Clinacanthus nutans* extracts against *Pseudomonas aeruginosa* on Mueller–Hinton agar. Acetone-extracted *Clinacanthus nutans* contained the highest level of antioxidant in comparison with other *Clinacanthus nutans* extracts.

## 1. Introduction

Wound healing, a skin repairing process, is one of the most complicated processes in the human body. To close an open wound, multiple cells and stages are intrinsically involved, including haemostasis, inflammation, angiogenesis, growth, re-epithelialization, and re-modelling. However, harmful substances or opportunistic microorganisms may invade the body and hamper this process.

According to [1], commensals, such as *Propionibacterium acnes*, *Streptococcus pyogenes*, and *Staphylococcus aureus*, can be easily isolated on the surface of the skin, whereas opportunists *Enterococcus* spp. and *Pseudomonas aeruginosa* can be isolated in the absence of healing and in the presence of persistent inflammation, eventually leading to chronic wounds. The existence of methicillin-resistant *Staphylococcus aureus* (MRSA) is one of the most challenging issues when dealing with chronic wounds, as colonization and replication of MRSA on the wound further delays the wound healing process [2]. With the discovery of antibiotics, bacterial infections were alleviated but at a dire cost with the emergence of antibiotic-resistant bacteria strains. These emerging strains were caused by the inappropriate usage of antibiotics in the past when healthcare systems lacked proper procedures for the identification of causative microbes [3]. Antimicrobial resistance is often associated with high morbidity and mortality. As a result, it has always been a worldwide bane. In 2000, a systematic review reported about proper wound care management, which included the use of antibiotics [4]. 

There are many species of anecdotal medical plants available in different regions of the world, where numerous indigenous peoples have utilized traditional medicine or herbal remedies for centuries. However, many of these are not systematically and scientifically documented as official medicines registered under the Food and Drug Administration (FDA, USA). This triggers the initiatives of scientists to analyze chemically and to delve deeper into bioactive phytochemicals and study their pharmacological properties. In order to bioprofile a plant that is efficacious in providing health benefits, there are a few properties required in the selection process, such as wound healing, antimicrobial, and antioxidant properties. 

*Clinacanthus nutans* (*Burm*.f.) *Lindau* (CN) is a well-known traditional herb used by natives in Southeast Asia, especially in Thailand and Malaysia. It is also colloquially known as Sabah snake grass in East Malaysia. In herbal remedies, the leaves are usually sun-dried and brewed as a tea. It is purportedly efficacious with the daily consumption of CN tea, which has led to cures in diabetes, dysuria, and diarrhea [5]. According to [6], CN is used to cure oxidative stress-related diseases and skin lesions in vivo caused by the *herpes simplex* virus and the *varicella-zoster* virus. In addition, some colon cancer survivors testified that the consumption of CN over time helped to ease chemotherapy effects and aided recovery from certain cancers. Correspondingly, CN was officially recognized as the principal medicinal plant for primary healthcare by the Ministry of Public Health in Thailand [7]. 

*Strobilanthes crispus* (L.) *Blume* (SC) originated from Madagascar and was slowly distributed to the Malay Archipelago region [8]. SC is a shrub with darker upper leaves than the underside, earning the name “Black-Faced General” by Chinese communities. The underside of the leaves is rough, whereas the upper surface is smooth, hence Malay communities named it “*Jin Batu*” or “*Pecah Beling*”. SC remedies have been reportedly used as antidiabetic, antilytic, laxative, anticancer, and diuretic agents. They have a high content of calcium carbonate, and, when boiled, the SC brew is slightly alkaline. It has been demonstrated to ease urination [9]. In Malaysia, natives chew and swallow fresh SC leaves to strengthen their immune system [10].

Previous studies have assessed the high antioxidant content and antimicrobial activities of CN and SC against some microbes. CN is commonly applied externally and available as a balm in the traditional market. SC is also reported to be capable of wound healing activities. A histological study on SC reported accelerated wound healing properties of SC ethanolic leaves extracts in rats [11]. However, there are only a few intensive studies on its wound healing properties. As such, this study aims to assess the wound healing properties, antimicrobial activities, and antioxidant properties of CN and SC.

This study also attempts to investigate the wound healing potential of different solvent extracts of CN and SC by using in vitro fibroblast-based scratch assays co-incubated with Methicillin-resistant *Staphylococcus aureus* (MRSA). 

## 2. Results and Discussion

### 2.1. Effects of CN and SC Extracts on Human Fibroblasts

In this study, a cell proliferation assay using 3-(4,5-Dimethylthiazol-2-yl)-2,5-Diphenyltetrazolium Bromide (MTT) was conducted on OUMS-36T-4F human fibroblasts in the absence and presence of extracts and bacteria as a baseline. 

The growth rate of MRSA is a critical parameter for the co-incubation assay. The growth of MRSA in the absence of extracts was determined and is shown in Figure 1. Before the MRSA was mixed into the culture media, the starting OD_600_ value was recorded. The OD_600_ values were between 0.27 and 0.50, corresponding to a different % of MRSA in the medium. The MRSA and fibroblasts were observed at 1, 3, 6, 10, 24, and 48 h intervals. The highest value recorded was 1% of MRSA in growth medium as observed at 24 h, which corresponded to 100% growth of MRSA. The findings showed that density of 1% MRSA achieved the highest growth at 24 h (Figure 1). The growth of different densities 0.25% and 0.1% of MRSA at the 24 h observation interval was 78.84% and 81.55%, respectively. Even though the 0.1% density of MRSA achieved a slightly higher growth compared to that of 0.25% density of MRSA, there was a significant difference with the 0.5% density of MRSA in the medium. As the growth of densities 0.25% and 0.1% of MRSA was similar, both were suitable to be used in the co-incubation assay. In this study, 0.1% MRSA was selected.

#### 2.1.1. Effects of CN Extracts on Human Fibroblasts

In the presence of CN extracts, at concentrations lower than 62.5 µg/mL, the proliferation of fibroblasts remained higher than 100% after incubating for 24 h (Figure 2). After incubation for 48 h in concentrations lower than 125 µg/mL, the cell viability remained higher than 100%, indicating that CN ethanol extract had little or no effect on the viability of the fibroblasts. Similarly, at concentrations of 62.50 µg/mL and lower for 24 h of incubation and at 125 µg/mL and lower for 48 h of incubation, the proliferation of fibroblast remained higher than 100% (Figure 3). For CN acetone extract, at concentrations of 7.81 µg/mL and lower for 24 h of incubation and at 31.25 µg/mL or lower for 48 h of incubation, the proliferation of fibroblasts was not affected, which is indicative that at higher concentrations, the extracts affected the proliferation of fibroblasts (Figure 4).

#### 2.1.2. Effects of SC Extracts on Human Fibroblasts

The results show that SC ethanol extract at concentrations of 62.50 µg/mL or lower did not affect the proliferation of fibroblasts after 24 h of incubation. On the other hand, after incubation for 48 h, the ethanol extract impeded the proliferation of the fibroblast with the exception of 3.91 µg/mL. For acetone and chloroform extracts, the concentrations at 62.5 µg/mL or lower for 24 h of incubation had little or no effect on the proliferation of the fibroblasts. For SC acetone extract, at 3.91 µg/mL after 48 h of incubation, only 90% of the fibroblasts were viable. When the fibroblasts wereincubated for 48 h in SC chloroform extract, the cell viability had dropped dramatically, even at 3.91 µg/mL.

Based on the MTT assay, SC extracts seemed to be more toxic than CN extracts when they were incubated for longer than 24 h (Figure 5, Figure 6 and Figure 7). In contrast to that of CN ethanol (Figure 2, Figure 3 and Figure 4), when the concentration of SC extracts was at 500 µg/mL, fibroblast cell viability was lower than 26% when incubated for 24 h and lower than 3% if incubated for 48 h (Figure 5, Figure 6 and Figure 7). Most of the cell viability of the fibroblasts was higher than 50% with CN extracts at concentrations of 500 µg/mL, and the lowest fibroblast cell viability recorded was CN acetone extract after incubating for 48 h. Both were extracted with acetone, and their fibroblast cell viability at the lowest concentration of 3.91 µg/mL was still lower compared to the other extracts from their respective plants (Figure 2 and Figure 5). Therefore, acetone extracts from plants are harmful to fibroblasts compared to ethanol and chloroform extracts. 

Based on the results shown in Figure 2, Figure 3 and Figure 4, the concentration of CN extract at 7 µg/mL enabled the fibroblast cell viability to remain at about 100% after incubating for 48 h, whereas the concentration was 70 µg/mL for all SC extracts. Thus, concentrations of 7 µg/mL for all CN extracts and 70 µg/mL for all SC extracts were selected to be used for the scratch assay.

#### 2.1.3. In Vitro Scratch Assay with and without Presence of MRSA

This study reports that the pseudo-wound of the control fully covered up after being incubated for 48 h (Table 1 and Table 2). However, when the CN ethanol and chloroform extracts were applied on the pseudo-wound, the fibroblasts closed the gap of the pseudo-wound after 24 h of incubation, indicative of a faster ‘healing’ rate. In contrast, the CN acetone extract only fully covered up the pseudo-wound after 48 h of incubation (Table 1). For SC extracts in Table 2, the pseudo-wound of the control fully covered up just after being incubated for 24 h. However, when only the SC chloroform extract was applied to the pseudo-wound, the pseudo-wound fully covered up after 48 h of incubation. The SC ethanol and acetone extracts did not aid in wound healing activity, but they slowed down wound healing activity and started to damage the fibroblasts after being incubated for 48 h. A time-lapse for the closing of the pseudo-wound with CN and SC extracts is shown in Figure 8 and Figure 9.

When the MRSA was involved in the co-incubation, the overall wound healing activity dropped. This fully represents the condition of an open wound when MRSA is invaded. All the co-incubation assay results showed a peak between 3, 6, or 10 h of incubation, and afterwards the results started to drop, which indicated that the MRSA started to attack the fibroblasts. The highest peak recorded for the CN extracts was chloroform extract at 52.4%, followed by 48.6% for acetone extract and then 43.1% for ethanol extract. For SC extracts, the highest peak recorded was 76.1% for ethanol extract, followed by chloroform and acetone extracts, whose peaks were recorded to be 60.2% and 45.8%, respectively. A time-lapse for the pseudo-wound with CN and SC extracts with MRSA present is shown in Figure 10 and Figure 11.

The results showed CN and SC extracts did not possess antimicrobial abilities and inhibition against MRSA (Table 3). Therefore, the CN and SC extracts did not inhibit the growth of MRSA in the flasks. CN extracts aided wound healing activity with low toxicity (Table 1), and SC extracts did not aid in the wound healing activity and contained higher toxicity compared to CN extracts, as shown by the MTT assay (Table 2). However, these CN and SC extracts were effective against *P. aeruginosa* (Table 3).

Due to the higher toxicity of SC extracts, there is a high possibility that the SC extracts inhibited MRSA activity and enabled the pseudo-wound to have time to regenerate, but eventually, when the amount of MRSA increased, MRSA took over the flask. Therefore, the highest peak recorded for the co-incubation was 76.1% for SC ethanol extract and followed by 60.2% for SC chloroform extract, which were both higher than the peak recorded from CN extracts. 

In conclusion, the evaluation of wound healing activity based on the MTT assay has proven that acetone extracts are toxic for fibroblasts and affect results at higher concentrations. When the CN ethanol and chloroform extracts were added to the pseudo-wound, the wound fully recovered after 24 h of incubation instead of after 48 h of incubation, as demonstrated by the control. However, when co-incubated with MRSA on the pseudo-wound, the MRSA took over the flasks after being incubated for 3 to 10 h, and it did not show any wound healing activity.

The SC extracts did not aid in the wound healing activity, but they further delayed the wound healing activity. However, when co-incubated with MRSA, the extracts inhibited MRSA activities, and the fibroblasts were able to regenerate for a short period before the MRSA took over after being incubated for 6 to 10 h.

### 2.2. Antimicrobial Activity

A disc diffusion assay was conducted on the CN and SC extracts to assess their antimicrobial activities. The CN and SC ethanol, acetone, and chloroform extracts were prepared at 1 mg/mL. According to the results from Table 3, only *P. aeruginosa* showed zones of inhibition in the disc diffusion assay. When CN ethanol, acetone, and chloroform extracts were applied on *P. aeruginosa*, the observed average zone of inhibition diameters were 15.7 mm, 14.3 mm, and 14 mm, respectively. When SC ethanol, acetone, and chloroform extracts were applied on the *P. aeruginosa*, the observed average zone of inhibition diameters were 14 mm, 12 mm, and 11 mm, respectively. According to [12], the minimum inhibitory concentrations (MIC) for CN ethanol and chloroform extracts against *P. aeruginosa* are 1.25 mg/mL and 0.63 mg/mL, respectively, whereas the MIC for *Staphylococcus aureus*, *E. coli*, and *K. pneumoniae* are 1.25 mg/mL and 2.5 mg/mL, 2.5 mg/mL or no antimicrobial activity, and 2.5 mg/mL and 2.5 mg/mL, respectively [12]. Based on Table 3, our results show the comparable significant antimicrobial properties of CN at 1 mg/mL against *P. aeruginosa*. However, no activity was detected in *S. aureus*, *E. coli*, and *K. pneumonia*. Similarly, antimicrobial activities were observed in SC ethanol, acetone, and chloroform extracts against *P. aeruginosa* but not in the other Gram-positive or Gram-negative bacteria.

In this study, CN and SC demonstrated antimicrobial activities against *P. aeruginosa* with CN extracts demonstrating higher efficacy against Gram-negative *P. aeruginosa* than SC extracts. Based on another study [13], most newly developed antibiotics are only active against Gram-positive bacteria due to the impermeable outer membrane of Gram-negative bacteria. However, this study showed that the antimicrobial properties of CN and SC extracts were efficacious against opportunistic Gram-negative *P. aeruginosa*. The components present in CN and SC might be targeting certain pathways or the DNA replication enzyme of the bacteria [14].

### 2.3. Chemical Analysis

Chemical analysis was performed to delineate the different phenolic and antioxidant properties of the extracts. This was conducted to tie in the observed antimicrobial activities detected in the disc diffusion assay. It was noted that the acetone extract had an antioxidant content about two times higher than that of the chloroform extract. This finding is supported by [15]. Acetone extract resulted in the highest antioxidant content, but ethanol with a higher polarity did not show a better result, which might be because most of the antioxidant components present in the CN were semi-polar or non-polar. Therefore, extraction solvent with semi-polarity such as acetone resulted in extract with higher antioxidant content compared to that of ethanol and chloroform as the extraction solvent. However, the types of antioxidants present within the extracts might differ. Therefore, further tests were involved to identify the types of antioxidants present in the extracts [16]. The results obtained from the SC extracts are different from those of the CN. The SC extracts have a higher green colour intensity compared to those of CN. A steep increasing curve can be observed, and there is a high possibility that the AAEAC value might be higher than 55% for all SC extracts. This finding is supported by [17], which shows that SC is a superior reactive oxygen species scavenger compared to CN. 

The findings clearly indicate that the total antioxidant content present in CN included phenols and polar flavonoids (Table 4). Additionally, ethanol and acetone are proven to be suitable and effective extraction solvents for phenolics and polar flavonoids extractions. However, both solvent extracts had different TPC and TFC values. The acetone extract had the highest antioxidant activity reported along with the highest TPC but was lower in TFC. Its total phenolic content was about two times higher than that of ethanol extract. On the other hand, the antioxidant activity of ethanol extract was slightly lower than that of the acetone extract, but it obtained the highest TFC of 0.472 mg RE/g defatted material. The TFC of acetone extract was reported to be about two times less than that of the ethanol extract. This result indicates that the antioxidant components present in the ethanol extract were mostly polar flavonoids and that the antioxidant components present in the acetone extract were mostly phenols. The total antioxidant content present from CN extracts ranged between 11.96% to 25% AAEAC, whereas the average total antioxidant content present SC extracts was mostly higher than 55%. Therefore, the total antioxidant content of SC was higher than that of CN. However, based on the results from TPC and TFC, the highest value of TPC recorded was 4.335 mg GAE/g defatted material from the CN acetone extract, whereas 0.472 mg RE/g defatted material was the highest value recorded for TFC from the CN ethanol extract.

## 3. Materials and Methods

### 3.1. Plants Collection

The studied plants, namely *Clinacanthus nutans* (CN) and *Strobilanthes crispus* (SC), were collected from two housing areas in Kuching, Sarawak. The leaves were cleaned and dried at room temperature for 3 weeks. The leaves were hand-picked and ground into powder using a domestic grinder (Sharp, Sakai, Japan). The powder was packed in a press-and-seal bag and kept in the dark for future use.

### 3.2. Extraction

There were three different solvents used in the extraction. These were ethanol (Analytical Reagent Grade, Fisher, Waltham, MA, USA), acetone (AR, Fisher, Waltham, MA, USA), and chloroform (AR, Fisher, Waltham, MA, USA). The cold maceration method was used for the extraction process. A sample of 50 g CN dried powder was weighed and mixed with 500 mL of ethanol. The mixture was left for 24 h with constant stirring with a magnetic stirrer (Favorit, Puchong, Malaysia). The mixture was then filtered with Whatman© No. 93 filter paper (Whatman©, USA). The solvent in the filtrate was removed using a rotary evaporator (BUCHI, Flawil, Switzerland) and left to dry completely in a fume hood. The extract with a concentration of 0.1 g/mL was transferred into a 50 mL centrifuge tube (Biologix, Jinan, China) and kept in a refrigerator at 2–8 °C in dark prior to the analysis.

The procedures were repeated with acetone and chloroform on the powdered samples of CN and SC, respectively.

### 3.3. Chemical Analysis

#### 3.3.1. Antioxidant 2,2-Diphenyl-1-picrylhydrazyl (DPPH) Assay

2,2-diphenyl-1-picrylhydrazyl (DPPH) (Sigma, St. Louis, MO, USA) free radical scavenging activity was assayed as described by [18,19] with slight modifications. The extracts went through a serial dilution with concentrations of 10, 20, 40, 60, 80, and 100 µg/mL with a final volume of 1 mL for each concentration. An amount of 1 mL of DPPH ethanolic solution (0.2 mM) was then added to each tube. The mixtures were sealed and swirled gently for 1 min then incubated in the dark for 30 min. The absorbance was read using a spectrophotometer at 517 nm. The values of DPPH radical scavenging were determined with a calibration curve as μg ascorbic acid (Sigma, St. Louis, MO, USA) equivalent per mL where ascorbic acid was used as the standard reference. All the extracts and standards were performed in triplicate (%).

The assay involved to determine the antioxidant content was DPPH radical scavenging, and the IC_50_ of ascorbic acid recorded from the standard curve was 5.5 µg/mL. It was used to calculate the Ascorbic Acid Equivalent Antioxidant Content (AAEAC) with the formula below:AAEAC = (IC_50_ of Standard)/(IC_50_ of Sample) × 100%

#### 3.3.2. Total Phenolic Content

The total phenolic content (TPC) of the plant extract was assessed with Folin–Ciocalteu Reagent as described by [19]. A total of 10 mg of crude extract was dissolved in each test tube with 1 mL of ethanol. After mixing, 200 µL of the mixture was made up to 3 mL. It was then followed by 0.5 mL of 50% (*v*/*v*) Folin–Ciocalteu Reagent and 2 mL of 20% (*w*/*v*) sodium bicarbonate solution. These were then incubated in the dark for 60 min at room temperature. The absorbance of the mixtures was measured at 650 nm with a spectrophotometer. Gallic Acid was used as a standard reference. All the samples for extracts and standards were conducted in triplicates.

#### 3.3.3. Total Flavonoids Content

The total flavonoid content (TFC) of the plant extract was determined by using the aluminium calorimetric method according to [20] with modification. First, 10 mg of the extract was dissolved in 1 mL of methanol. After the extract was fully dissolved in the methanol, 500 µL was transferred into another tube and up to 5 mL. After mixing, 1 mL of the mixture was transferred into another tube with 1 mL of 2% (*w*/*v*) AlCl_3_ and incubated in the dark for 10 min. The absorbance was measured with a spectrophotometer at 420 nm. Rutin was used as the standard reference for this test. All the extracts and standards were carried out in triplicates.

### 3.4. Disc Diffusion Assay

A total of eight bacteria species, encompassing Gram-positive and Gram-negative species commonly present in our surroundings and known opportunistic pathogens, were used in this study. The Gram-positive bacteria involved were *Streptococcus pyogenes* (ATCC No. 25663, Manassas, VA, USA), *Staphylococcus aureus* (ATCC No. 25923, Manassas, VA, USA), and Methicillin-resistant *Staphylococcus aureus* (ATCC No. 33591, USA). The Gram-negative bacteria involved were *Pseudomonas aeruginosa* (ATCC No. 109246, Manassas, VA, USA), *Escherichia coli* (ATCC No. 1100101, USA), *Shigella species* (ATCC No. 23354, Manassas, VA, USA), *Salmonella typhimurium* (ATCC No. 14028, USA), and *Klebsiella pneumonia* (ATCC No. BAA-1705, Manassas, VA, USA). Nutrient broth (NB) (Oxoid, Basingstoke, UK) was used for bacterial growth. The bacteria were freshly cultured 24 h before it was plated on Mueller–Hinton Agar (MHA) (Oxoid, Basingstoke, UK) to perform the agar disc diffusion test.

A total of 1 L distilled water was poured into a 1 L Scott bottle containing 13 g of NB powder. The NB was stirred and heated util it dissolved before being aliquoted into a 3 mL Bijou bottle (Biologix, Jinan, China) and autoclaved at 121 °C for 15 min. The aliquots of NB were stored at 4 °C for future use. MHA was made by dissolving 38 g of MHA powder in 1 L of distilled water in a 1 L glass bottle and autoclaved at 121 °C for 15 min. The MHA was cooled in a 60 °C water bath and poured into a Petri dish (Favorit, Puchong, Malaysia). The plates of MHA were left to cool and solidify before storage at 4 °C for future use.

The disc diffusion method was performed as described by [21] with slight modification. The bacteria were spreading uniformly on a sterile MHA plate. The extracts were prepared at a concentration of 1 mg/mL. Sterile empty discs (Oxoid, Basingstoke, UK) were placed at the designed area on the plate, and 20 µL of the respective extract was pipetted onto the sterile disc. The procedures were repeated for all bacteria in triplicates. A positive control Levofloxacin 5 mg disc (Oxoid, Basingstoke, UK) was placed along with negative control, MH broth (Oxoid, Basingstoke, UK) on another agar plate with triplicates. The plates were incubated at 37 °C and were observed at 24, 48, and 72 h. The zone of inhibition diameter for each extract was observed, measured in mm, and recorded.

### 3.5. Cell Viability of Skin Fibroblast with CN and SC Extracts

Skin fibroblast (OUMS-36T-4F) was selected for the in vitro scratch assays. The cells were cultured with Dulbecco’s modified Eagle’s medium (DMEM) (Gibco, Waltham, WA, USA) with 10% Fetal Bovine Serum (FBS) (Gibco, Waltham, WA, USA). Before determination of the Minimum Inhibition Concentration (MIC) and bacterial selection, a scratch assay without extracts and bacteria was performed on human fibroblasts to determine the observation time for the wound healing activity. 

The scratch assay was performed according to [22] with modifications on the total volume of media added. Skin fibroblasts were cultured in a T25 flask (Biologix, Jinan, China) overnight. When the cells attached to the surface and reached a confluency of 80%, the culture media were removed, and a 1 mL sterile pipette tip was used to scratch on the marked area to form a pseudo-wound. Then, 2 mL culture media was added to the T25. The cells were incubated at 37 °C and 5% CO_2_ and were observed under the microscope at intervals of 1, 3, 6, 10, 24, and 48 h.

This study demonstrates that, despite observations of up to 48 h, 24 h of incubation is sufficient for the cells to migrate and cover up the pseudo-wound. The scratch assay was repeated with the addition of extracts into the culture media. The 3-(4,5-dimethylthiazol-2-yl)-2,5-diphenyltetrazolium bromide (MTT) assay was conducted to determine the Minimum Inhibition Concentration (MIC) of each extract when exposed to the human fibroblast. A working solution of 10% MTT in phosphate-buffered saline (PBS) was reconstituted for this assay.

Cell optimization was performed to observe the conditions of the cells after 24 and 48 h of incubation. The densities of the cells started with 10,000 cells were serial diluted to 6250 cells. They were incubated at 37 °C and 5% CO_2_ for 24 h. After 24 h of incubation, the media were replaced with new media and 20 µL of 10% MTT reagent. The mixture was incubated in the dark for 3 h before the formazan crystals were solubilized with 100 µL Dimethyl sulfoxide (DMSO) (Sigma, Rowville, Australia) and mixed thoroughly. The result was read at 570 nm by using a microplate reader (SpectraMax iD3 Multi-Mode Microplate Reader, Molecular Devices, San Jose, CA, USA). The cell viability was calculated according to the formula below:Cell viability (%) = (Absorbance of sample)/(Absorbance of control) × 100%

According to the result obtained, a concentration of 2.2 × 10^4^ cells was selected for the MTT assay to determine the MIC of the extracts. Firstly, 2.2 × 10^4^ cells were seeded in 96-well flat-bottom plates (Biologix, Jinan, China) for 24 h. The extracts were dissolved in the culture media and filtered with a 0.45 µm hydrophilic filter (Sartorius, Göttingen, Germany). The culture media were replaced by the extracts, starting with a concentration of 500 µg/mL and 1:2 serial dilutions to 3.906 µg/mL. The treated cells were incubated for 24 and 48 h at 37 °C and 5% CO_2_. After 24 h of incubation, the extracts were replaced with 200 µL fresh culture media and 20 µL of 5 mg/mL 10% MTT reagent. These cells were incubated in the dark for 3 h, and 100 µL DMSO was added and mixed thoroughly to solubilize the formazan crystals. The absorbance was measured at a wavelength of 570 nm using a microplate reader (SpectraMax iD3 Multi-Mode Microplate Reader, Molecular Devices, San Jose, CA, USA).

#### In Vitro Scratch Assay with and without Presence of MRSA

Methicillin-resistant *Staphylococcus aureus* (MRSA) was selected to co-incubate in this in vitro assay, as it is widely recognized for its persistent virulence when treating open wounds. Based on the observation from the scratch assay, it took the pseudo-wound up to 24 h to ‘heal’. A 10 µL inoculation loop was used to transfer 10 µL of MRSA into an aliquot of 3 mL fresh NB and incubated overnight. Before the MRSA was transferred into the culture media, the OD_600_ of the MRSA was read and recorded using a spectrophotometer (DeNovix DS-11 FX, Wilmington, NC, USA). 

The different volumes of MRSA that were added were 100 µL (5%), 20 µL (1%), 10 µL (0.5%), 5 µL (0.25%), and 2 µL (0.1%), and they comprised a final volume of 2 mL (Table 4). They were incubated at 37 °C and 5% CO_2_. The observations were recorded immediately and after incubation of 1, 3, 6, 10, 24, and 48 h. In the MTT assay, the extracts with different solvents exhibited different MIC. Concentrations for CN at 7 µg/mL and SC at 70 µg/mL were used. When different volumes of MRSA were applied in the scratch assay, the OD_600_ of the MRSA obtained had to fall between 0.4 to 0.5 to obtain the best result for a 48 h assay. With 2 µL of MRSA added, this was indicative that only 0.1% of the total media was applied to the cells. 

The scratch assay was performed according to [21] with modifications on the total volume added along with the co-incubation of MRSA. The human fibroblasts were cultured in a T25 flask and incubated for 24 h to ensure the adherence of cells with a confluency of 80% at 37 °C and 5% CO_2_. A pseudo-wound was etched on the fibroblasts with a sterile 1 mL pipette tip on the marked area. The media with CN and SC extracts were co-incubated with the pseudo-wound, and they were observed and recorded (Nikon Ti2 Inverted Microscope, Tokyo, Japan) immediately post scratch and after incubation for 1, 3, 6, 10, 24, and 48 h.

The same procedure was repeated with the addition of MRSA into the media with CN and SC extracts and co-incubated onto a new set of pseudo-wounds. These pseudo-wounds were observed and recorded at the same durations as stated above. 

### 3.6. Statistical Analyses

All results were expressed as the mean of three replicates ± standard deviation. The results of the scratch assay in triplicates were analysed with ImageJ and expressed in % of cells in the area.

## 4. Conclusions

The findings of the scratch assay show that CN extracts perform better than SC in aiding the wound healing activity. However, when co-incubation with MRSA is involved in the scratch assay, SC performs better in aiding the wound healing activity. In conclusion, CN exhibits wound healing properties and performs better than SC. However, when MRSA is present in the wound, SC is superior in resisting the growth of MRSA, and it allows the wound to heal better before the MRSA takes over.

These findings delineate the superior of these two herbal extracts in their abilities to promote wound healing in the presence of a medically significant bacteria that is known to inflict morbidity and mortality on patients, particularly on those who are immuno-compromised. This study also documents the efficacies of these traditional herbal remedy plants in a systematic and scientific manner. These results enable the build-up of a potentially extensive database for traditional herbal plants. In addition, these outcomes have revealed several knowledge gaps worthy of further investigations, including the mechanism of action of these extracts against MRSA and its biofilm.

## Figures and Tables

**Figure 1 molecules-27-01722-f001:**
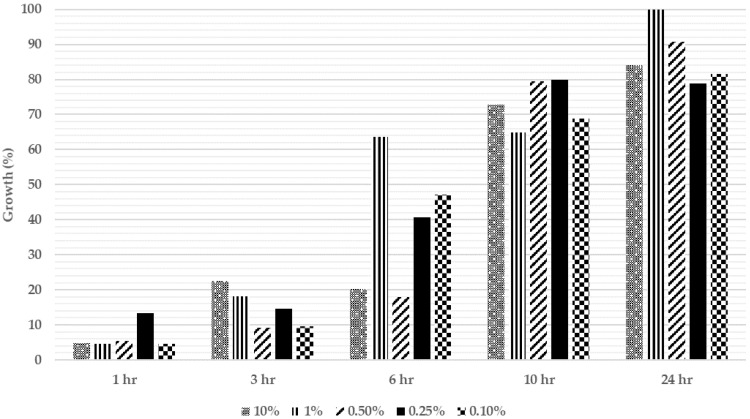
Growth of different densities of MRSA (%) over 24 h when co-incubated with human fibroblasts in a growth medium only.

**Figure 2 molecules-27-01722-f002:**
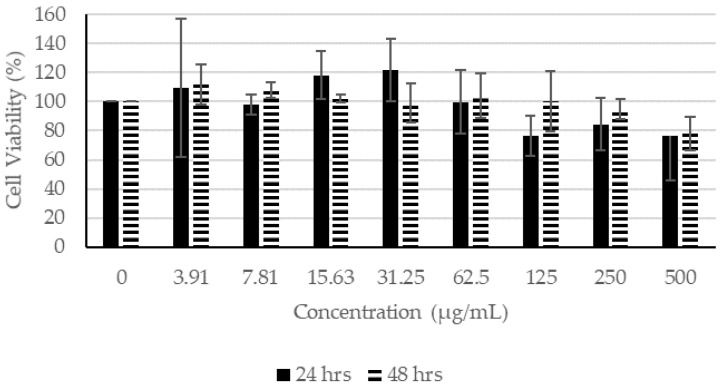
Cell Proliferation Assay. The human fibroblasts were incubated with different concentrations of CN ethanol extract for 24 and 48 h.

**Figure 3 molecules-27-01722-f003:**
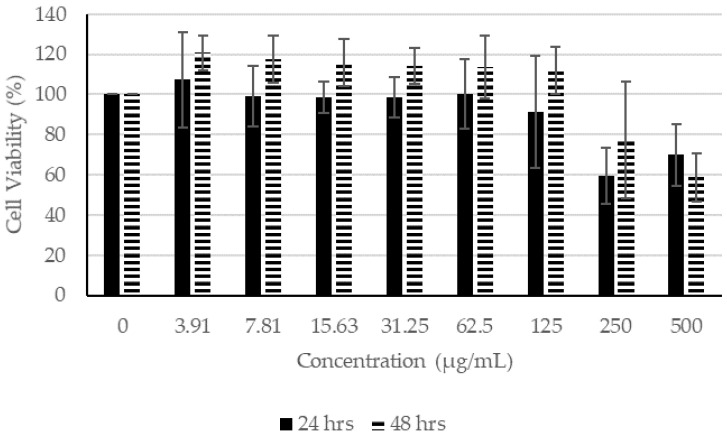
Cell Proliferation Assay. The human fibroblasts were incubated with different concentrations of CN chloroform extract for 24 and 48 h.

**Figure 4 molecules-27-01722-f004:**
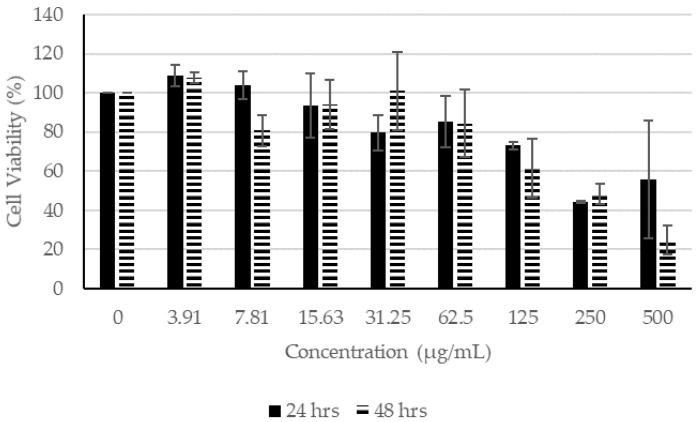
Cell Proliferation Assay. The human fibroblasts were incubated with different concentrations of CN acetone extract for 24 and 48 h.

**Figure 5 molecules-27-01722-f005:**
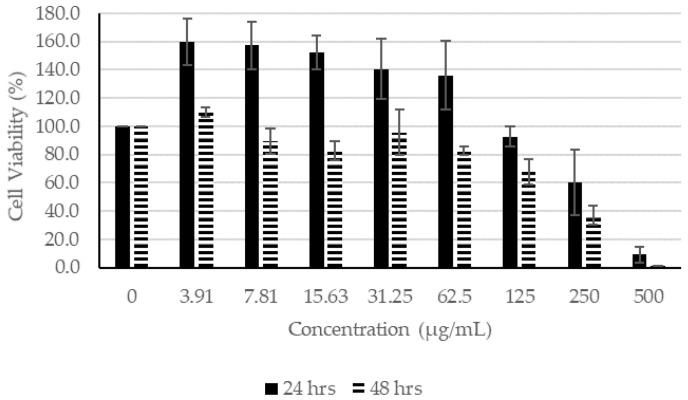
Cell Proliferation Assay. The human fibroblasts were incubated with different concentrations of SC ethanol extract for 24 and 48 h.

**Figure 6 molecules-27-01722-f006:**
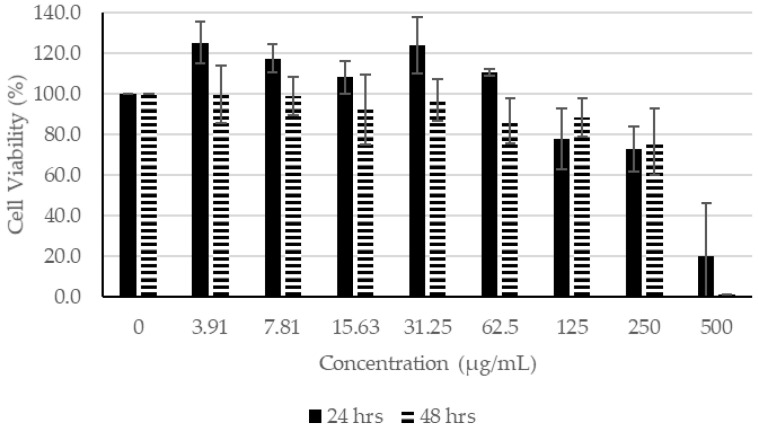
Cell Proliferation Assay. The human fibroblasts were incubated with different concentrations of SC chloroform extract for 24 and 48 h.

**Figure 7 molecules-27-01722-f007:**
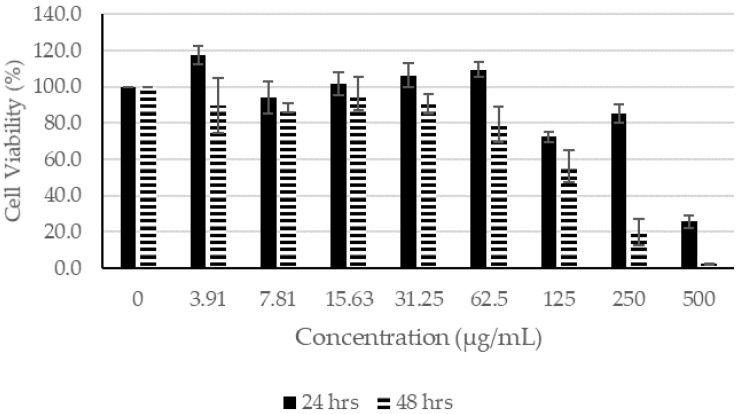
Cell Proliferation Assay. The human fibroblasts were incubated with different concentrations of SC acetone extract for 24 and 48 h.

**Figure 8 molecules-27-01722-f008:**
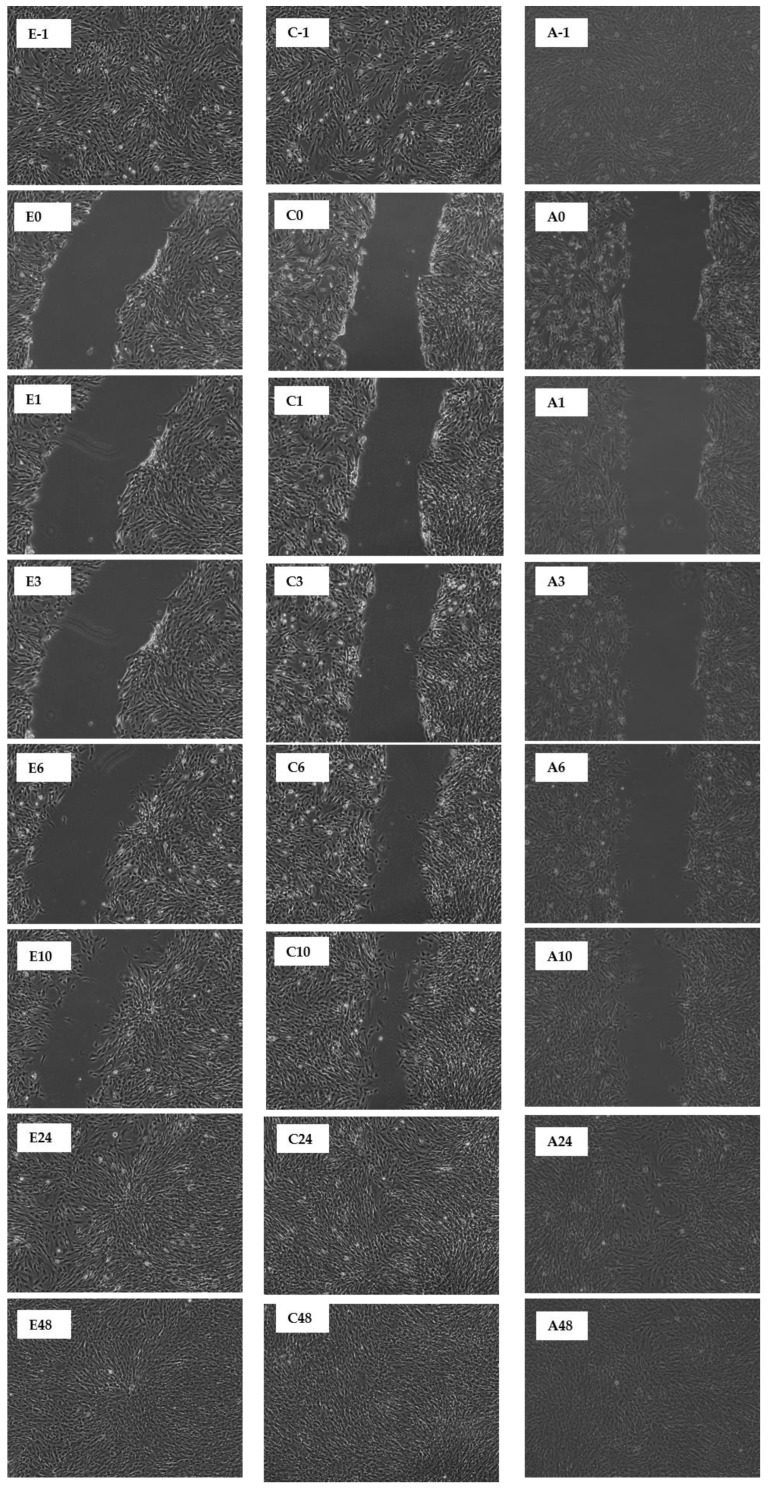
Scratch Assay on human fibroblasts in the presence of CN extracts (7 µg/mL). An in vitro wound healing process was observed in a time-dependent manner on human fibroblasts in the presence of ethanol (E), chloroform (C), and acetone (A). The durations are denoted as follows: -1: before scratch; 0: immediately post scratch; 1: 1 h post scratch; 3: 3 h post scratch; 6: 6 h post scratch; 10: 10 h post scratch; 24: 24 h post scratch; 48: 48 h post scratch.

**Figure 9 molecules-27-01722-f009:**
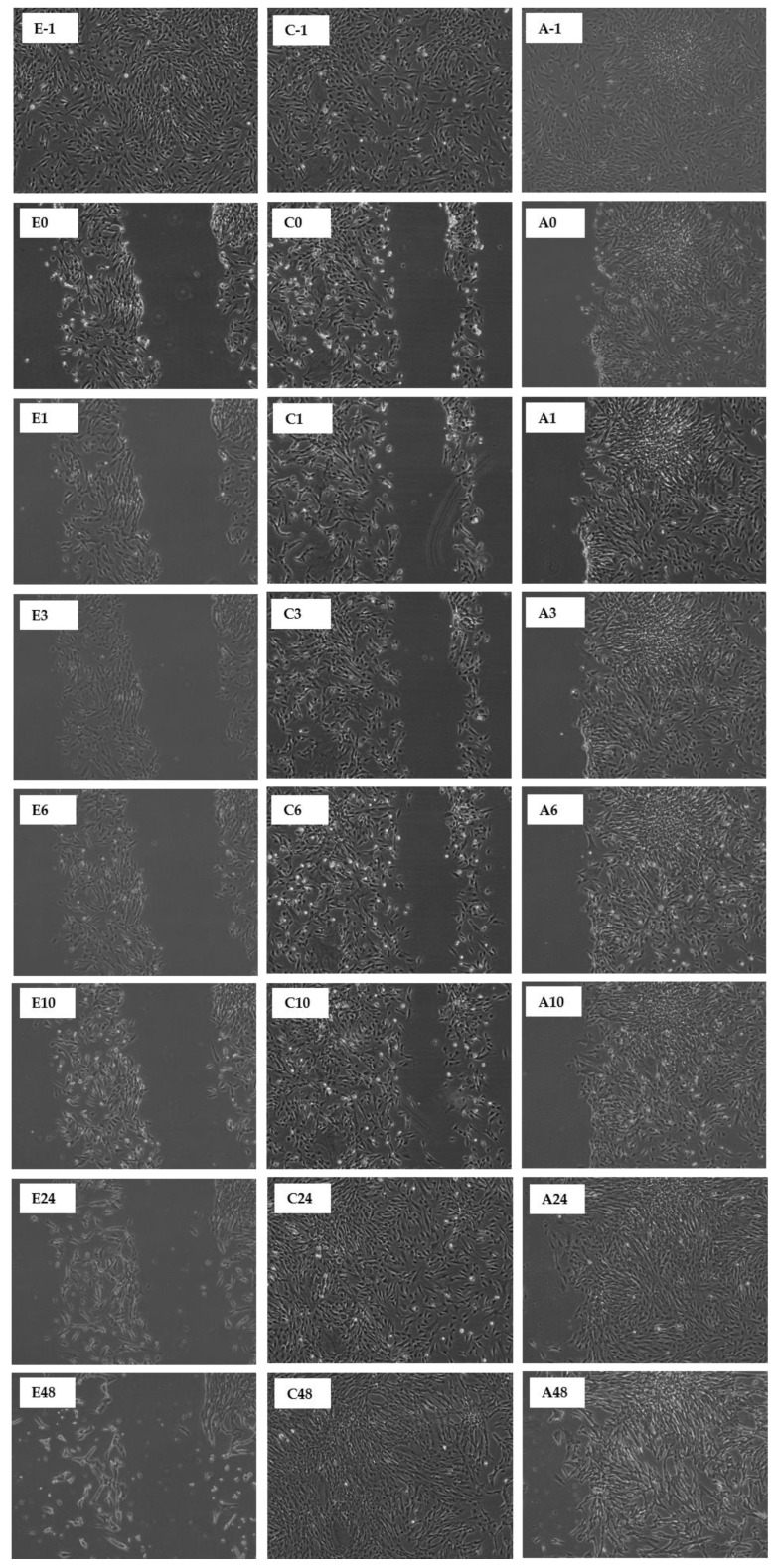
Scratch Assay on human fibroblasts in the presence of SC extracts (70 µg/mL). An in vitro wound healing process was observed in a time-dependent manner on human fibroblasts in the presence of ethanol (E), chloroform (C), and acetone (A). The durations are denoted as follows: -1: before scratch; 0: immediately post scratch; 1: 1 h post scratch; 3: 3 h post scratch; 6: 6 h post scratch; 10: 10 h post scratch; 24: 24 h post scratch; 48: 48 h post scratch.

**Figure 10 molecules-27-01722-f010:**
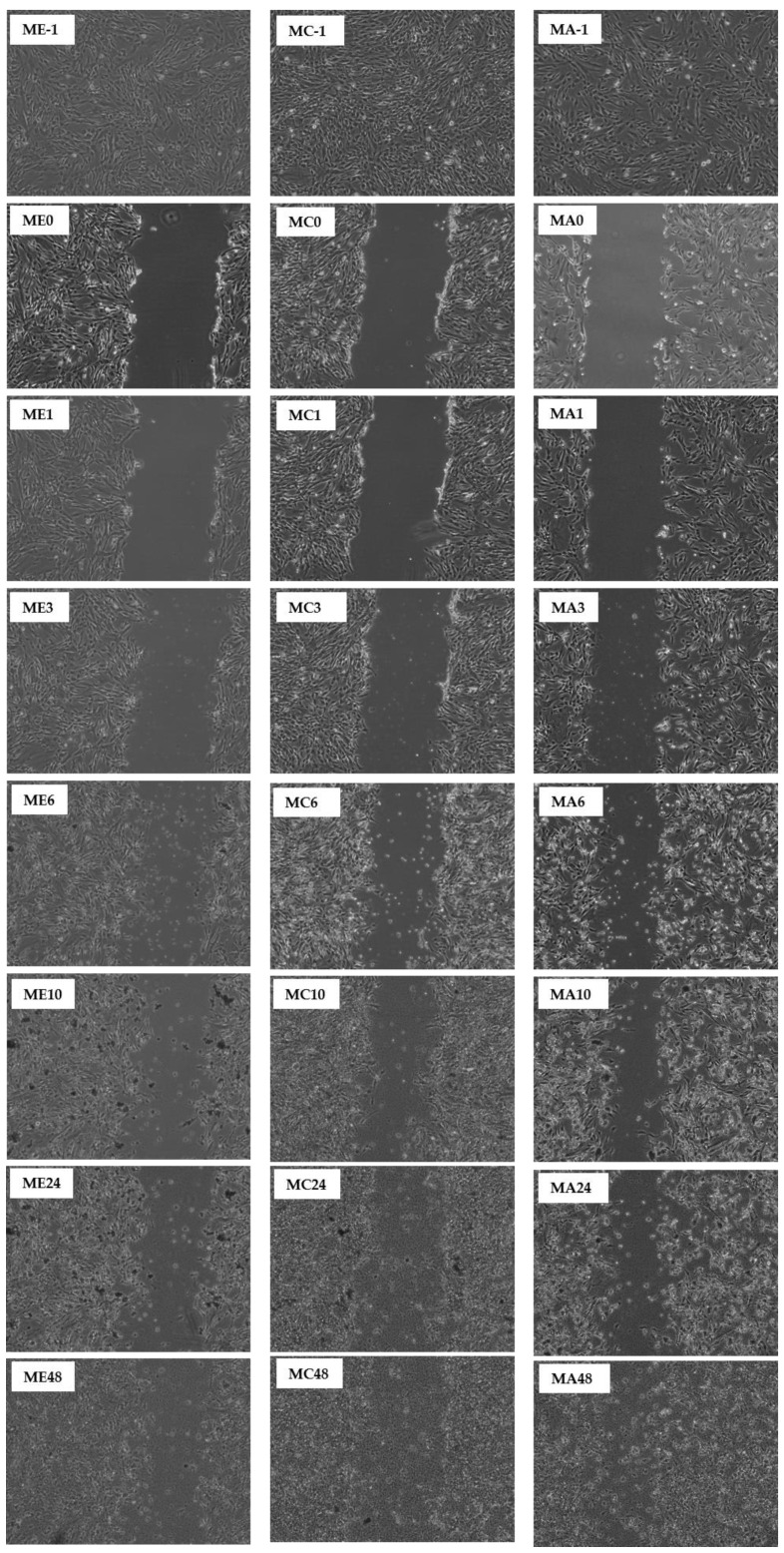
Scratch Assay on human fibroblasts in the presence of CN extracts (7 µg/mL) and MRSA. An in vitro wound healing process was observed in a time-dependent manner on human fibroblasts in the presence of ethanol (E), chloroform (C), and acetone (A). The durations are denoted as follows: -1: before scratch; 0: immediately post scratch; 1: 1 h post scratch; 3: 3 h post scratch; 6: 6 h post scratch; 10: 10 h post scratch; 24: 24 h post scratch; 48: 48 h post scratch.

**Figure 11 molecules-27-01722-f011:**
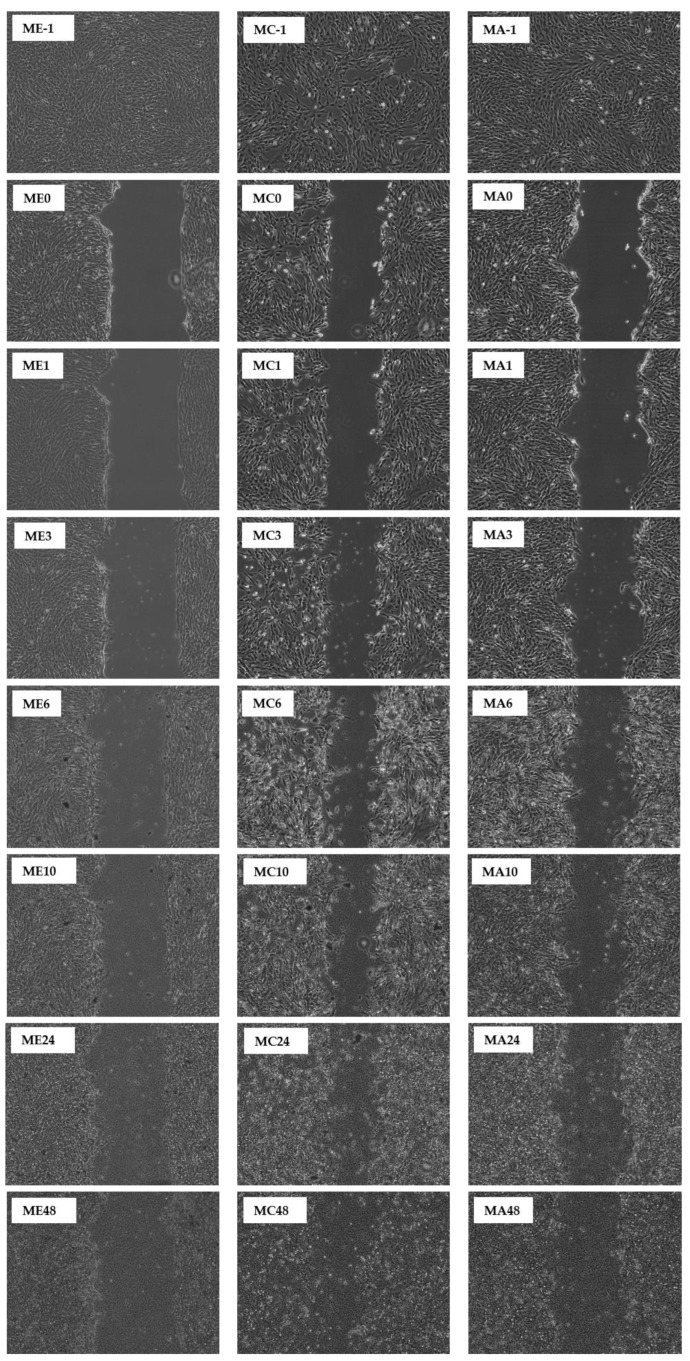
Scratch Assay on human fibroblasts in the presence of SC extracts (70 µg/mL) and MRSA. An in vitro wound healing process was observed in a time-dependent manner on human fibroblasts in the presence of ethanol (E), chloroform (C), and acetone (A). The durations are denoted as follows: -1: before scratch; 0: immediately post scratch; 1: 1 h post scratch; 3: 3 h post scratch; 6: 6 h post scratch; 10: 10 h post scratch; 24: 24 h post scratch; 48: 48 h post scratch.

**Table 1 molecules-27-01722-t001:** In vitro Scratch Assay. Effects of CN extracts (7 µg/mL) co-incubated with fibroblasts in the absence and presence of MRSA in dose-dependent and time-dependent assays, respectively. The results are expressed in the % of the number of cells in that area.

Incubation Time (h)		0	1	3	6	10	24	48
	Control	36.8	40.2	44.9	58.0	70.4	85.1	113.4
Extract only	Ethanol	41.2	47.0	56.7	61.8	71.9	114.7	141.3
	Chloroform	41.7	52.2	56.4	59.5	74.1	111.1	137.1
	Acetone	22.9	26.1	25.1	30.3	39.7	83.6	100.9
Co-incubation with extract and MRSA	Ethanol	28.1	30.9	38.3	43.1	38.0	39.2	33.7
	Chloroform	35.0	36.3	37.9	44.0	52.4	47.7	46.2
	Acetone	36.3	38.5	48.6	44.0	41.7	37.2	20.8

**Table 2 molecules-27-01722-t002:** In vitro Scratch Assay. Effects of SC extracts (70 µg/mL) co-incubated with fibroblasts in the absence and presence of MRSA in dose-dependent and time-dependent assays, respectively. The results are expressed in the % of the number of cells in that area.

Incubation Time (h)		0	1	3	6	10	24	48
	Control	47.4	54.8	60.2	79.8	94.9	102.7	137.9
Extract only	Ethanol	32.9	35.9	42.0	43.5	42.5	43.5	35.7
	Chloroform	26.0	26.1	32.7	38.6	52.4	78.3	101.0
	Acetone	39.3	40.8	45.3	51.0	53.1	73.0	62.3
Co-incubation with extract and MRSA	Ethanol	42.0	52.9	59.0	64.5	76.1	58.4	13.4
	Chloroform	29.4	37.3	43.0	57.7	60.2	42.2	25.7
	Acetone	28.7	34.3	40.5	45.8	31.1	20.1	12.5

**Table 3 molecules-27-01722-t003:** Antimicrobial potential of ethanol, acetone, and chloroform extracts for both CN and SC at 1 mg/mL. This table shows the zone of inhibition (mm) when different CN and SC extracts at 1 mg/mL are treated on Gram-positive and Gram-negative bacteria on Mueller–Hinton Agar.

	*Clinacanthus nutans* (CN)									*Strobilanthes crispus* (SC)									Control		
Extract	Ethanol			Acetone			Chloroform			Ethanol			Acetone			Chloroform			Levofloxacin (5 mg)		
Duration of exposure (h)	24	48	72	24	48	72	24	48	72	24	48	72	24	48	72	24	48	72	24	48	72
Gram-positve bacteria																					
*S. pyogenes*	X	X	X	X	X	X	X	X	X	X	X	X	X	X	X	X	X	X	43	42	43
*S. aureus*	X	X	X	X	X	X	X	X	X	X	X	X	X	X	X	X	X	X	10	11	11
MRSA	X	X	X	X	X	X	X	X	X	X	X	X	X	X	X	X	X	X	7	X	X
Gram-negative bacteria																					
*E. coli*	X	X	X	X	X	X	X	X	X	X	X	X	X	X	X	X	X	X	42	41	39
*P. aeruginosa*	15	16	16	14	14	15	14	14	14	15	14	13	12	12	12	11	11	11	27	25	24
*S. typhimurium*	X	X	X	X	X	X	X	X	X	X	X	X	X	X	X	X	X	X	30	31	31
*Shigella* sp.	X	X	X	X	X	X	X	X	X	X	X	X	X	X	X	X	X	X	30	33	34
*K. pneumoniae*	X	X	X	X	X	X	X	X	X	X	X	X	X	X	X	X	X	X	24	28	27

**Table 4 molecules-27-01722-t004:** The comparison of the antioxidant content (AAEAC), total phenolic content (TPC), and total flavonoids content (TFC) between CN and SC with different extraction solvents.

	*Clinacanthus nutans* (CN)			*Strobilanthes crispus* (SC)		
	Ethanol	Acetone	Chloroform	Ethanol	Acetone	Chloroform
AAEAC (%)	17.74	25.00	11.96	>55	>55	>55
TPC (mg GAE/g defatted material)	0.258	4.335	2.934	1.878	0.165	0.207
TFC (mg RE/g defatted material)	0.472	0.190	0.080	0.2524	0.2777	0.3127

Note: Ascorbic Acid Equivalent Antioxidant Content = AAEAC; TPC recorded as mg Gallic Acid Equivalent/g defatted material; TFC recorded as mg Rutin Equivalent/g defatted material.

## Data Availability

Not applicable.

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
