# Peer review of "Wound Healing, Antimicrobial and Antioxidant Properties of Clinacanthus nutans (Burm.f.) Lindau and Strobilanthes crispus (L.) Blume Extracts"

_molecules, 2022, doi:10.3390/molecules27051722_

Round 1

Reviewer 1 Report

This manuscript « Wound Healing, Antimicrobial and Antioxidant Properties of Clinacanthus nutans (Burm.f.) Lindau and Strobilanthes crispus (L.) Blume extracts » is submitted has an article in Molecules.

This manuscript does not present the quality to be published in Molecules.

Several major mistakes

Table 1 and 2 the part « Co-incubation with extract and MRSA » are identical

No details of the extraction procedure are given, how do you determine the concentration of the extracts ? L259 authors write about 7 µg/mL and 70 µg/mL in order to standardize the concentration. Later assays were performed with 10 mg/mL or 200 mg/mL. The concentrations of the extracts are not given in the figure legends.

The antimicrobial activities are not clear is there efficiency against P. aureus no values in Table 3, but « However, previous antimicrobial studies on CN showed positive results when it was applied to B. cereus, Staph. aureus, E. coli, and Kleb. pneumoniae as well [12] ». However there is no related data in this article.

Section 2.3 to 2.5 are relevant to materials and methods

The « second » Table 4 is not necessary and what are DMEM and FBS ?

Globally, the manuscript need to be improved, analysis of the data in the results is quite poor and the discussion section is very short.

Reviewer 2 Report

  1. The English need improvement since there are some grammatical and syntax errors in the manuscript. For example,
  • in line number 83, the word “balm” may be as “a balm”;
  • in line number 95, “baseline” as “a baseline”;
  • in line number 128 and 170, “number” as “the number”;
  • in line number 230, “zone” as “a zone”;
  • in line number 232, “of indistinguishable” as “an indistinguishable”;
  • in line number 234, “clear” as “a clear”;
  • in line number 305, “because” as “be because”;
  • in line number 306, “when extraction” as “extraction”;
  • in line number 307, “acetone were” as “acetone was”;
  • in line number 314, “SC have” as “SC has”;
  • in line number 315, “that the” as “the”;
  • in line number 325, “acetone” as “the acetone”;
  • in line number 396, “extract were” as “extract was”;
  • in line number 473, “scratch” as “the scratch”.

The grammar mistakes which are not mentioned here are also to be checked and corrected properly.

  1. There are some typing mistakes as well, and authors are advised to carefully proof-read the text. For example,
  • in line number 19, the words “Gram positive” may be as “Gram-positive”;
  • in line number 59, “consuming” as “consumption”;
  • in line number 111, “different density” as “different densities”;
  • in line number 125, “time-lapsed” as “time-lapse”;
  • in line number 415, “3 , 6 , 10 , 24 , a” as “3, 6, 10, 24,”;
  • in line number 421, “phosphate buffered” as “phosphate-buffered”;
  • in line number 430 an 443, “Multi- Mode” as “Multi-Mode”;
  • in line number 435, “flat bottom” as “flat-bottom”.

The typos not mentioned here are also to be checked and corrected properly.

  1. Check the abbreviations throughout the manuscript and introduce the abbreviation when the full word appears the first time in the text and then use only the abbreviation (For example, Clinacanthus nutans (CN), Strobilanthes crispus (SC), methicillin-resistant Staph. aureus (MRSA), etc.,). And it should be in both abstract as well as in the remaining part of the manuscript. Make a word abbreviated in the article that is repeated at least three times in the text, not all words need to be abbreviated.
  2. The full form of the species should be given when the first time appears and followed by only the first letter of the genus in both the abstract and the remaining part of the manuscript (For example, Pseudomonas aeruginosa / Escherichia coli / Staphylococcus aureus when the first time appears and followed by P. aeruginosa / E. coli / S. aureus). 
  3. The author should first mention in the manuscript, binomial classification must be with the family of the plant used in the present study and in the introduction section.
  4. In the results, figure numbers 5, 9, 10 and 11 may be given with colour images for a better understanding of the results.
  5. The reference cited in the results section should be removed and it may be given in any other part of the manuscript (For example, in line number 237, 242, 254 etc., ). The results should be key points of the overall observation of the present study only and not with others.
  6. The discussion is too simple. The authors may improve the discussion of their work by focusing on the present findings and introducing other authors who also worked with the same or other studies with recent references shortly and possibly with the parameters concentrated in the present findings.
  7. The authentication of the plant may be included in the materials and methods, if possible with voucher numbers. The technical terms (Latin Phrase) “in vitro (in line number 205) should be italic and it should be checked all over the manuscript.
  8. In statistical analysis, the software is not mentioned in the materials and methods. Therefore, it has been strongly recommended to include the software (with its version) under the heading statistical analysis.

Reviewer 3 Report

Thanks for the opportunity to review this research. The manuscript entitled „Wound Healing, Antimicrobial and Antioxidant Properties of Clinacanthus nutans (Burm.f.) Lindau and Strobilanthes crispus (L.) Blume extracts” have described the wound healing potential of different solvent extracts of CN and SC respectively by using in vitro fibroblast-based scratch assay co-incubated with Methicillin-resistant Staphylococcus aureus. The subject of the manuscript is topical, but I recommend the publishing of the paper after the necessary corrections.

  1. There are several typographical mistakes as well in whole manuscript. Therefore, the author’s thoroughly careful check the language and typo mistake to minimize the error.
  2. The abstract should be beginning with a sentence about the background of concept and the aims as well as novelty of study should be mentions. What exactly is the novelty of this study? Abbreviations must be avoided in abstract. Parenthesis should be avoided in abstract. Please improve.
  3. Introduction; Check and format the citations in the whole manuscript. Also, Appropriate references must be provided to explained the background, what is already done and why this study carried out. Other vise the novelty of this research is still poorly presented. This is important especially for the high IF journals. The scientific style should be used. What exactly is the aim of this work? Hypothesis statement is missing in the introduction section.
  4. Material and methods; The whole M&M section must be substantially rewritten and improved. The methods are not properly referenced and are not possible to follow, reproduce and verify.
  5. Results and discussion; General remark to the discussion - In my opinion, the discussion provided by Authors is difficult to follow and verify due missing critical details in the methodology section.

Round 2

Reviewer 1 Report

The revised version of the manuscript is again submitted as an article.

 The improvement is very poor

Also you even not apologized for massive mistakes such wrong reference and double Table 4

The concentrations of the extracts should be given in the figure legends.

The sections 2.2 and 2.3 are just introduced as results without any discussion or relevance with the other results. There is no interpretation just crude data and comparison.

The table 3 and the multiple sentences that extracts have antimicrobial activities should be removed. The only evidence for activity is SC extracts and P. aeruginosa, you can only assume CN extracts are efficient based on ref 12. The Table 3 is without interest because you used low concentration as written L 309.

Also in the Table 3 legend is not clear.

Author Response

We would like to express our thanks to the reviewer for the time and effort in this manuscript. We apologise for the mistakes present and endeavour to rectify these to the satisfaction of the reviewer.

Reviewer Comment No 1

The revised version of the manuscript is again submitted as an article.

Author Feedback 1

We have proofread the manuscript once more. We are now submitting it for another round of review as an article.

Reviewer Comment No 2

The improvement is very poor.

Author Feedback 2

We apologised on the very poor improvement. We endeavour and hope we had rectified the manuscript as advised to your satisfaction.

Reviewer Comment No 3

Also you even not apologized for massive mistakes such wrong reference and double Table 4.

Author Feedback 3

We apologised for these oversight. The duplicated table 4 had been removed. The reference had been corrected.

Reviewer Comment No 4

The concentrations of the extracts should be given in the figure legends.

Author Feedback 4

We had included the concentrations of the CN extract (7 µg/mL) and SC extracts (70 µg/mL) respectively in the figure legends of figures 8-11.

Reviewer Comment No 5

The sections 2.2 and 2.3 are just introduced as results without any discussion or relevance with the other results. There is no interpretation just crude data and comparison.

Author Feedback 5

We had included a brief description and purpose for the use of disc diffusion assay (2.2) and chemical analysis (2.3) respectively to tie in with the results obtained.

Reviewer Comment No 6

The table 3 and the multiple sentences that extracts have antimicrobial activities should be removed. The only evidence for activity is SC extracts and P. aeruginosa, you can only assume CN extracts are efficient based on ref 12. The Table 3 is without interest because you used low concentration as written L 309.

Author Feedback 6

For this study, table 3 was used to depict that tests were done on several strains of bacteria known to have acquire antibiotic resistance. Yet, the extracts were solely efficacious against one opportunistic Gram negative P. aeruginosa. The data in table 3 were used as comparison with study done in [12]. Despite a low concentration of 1 mg/mL, the study showed that it was comparable to other studies recorded in [12]. With careful deliberation, the author hopes the reviewer would allow table 3 to remain.

Reviewer Comment No 7

Also in the Table 3 legend is not clear.

Author Feedback 7

We had added more specific details in the figure legend to address the reviewer comment.

Once again, we are grateful for your time and guidance. Thank you.
